# SInGE: Sparsity via Integrated Gradients Estimation of Neuron Relevance

**Edouard Yvinec**[1,2] , **Arnaud Dapogny**[2] , **Matthieu Cord**[1] , **Kevin Bailly**[1,2]

Sorbonne Université[1], CNRS, ISIR, f-75005, 4 Place Jussieu 75005 Paris, France
Datakalab[2], 114 boulevard Malesherbes, 75017 Paris, France
`ey@datakalab.com`

## Abstract

The leap in performance in state-of-the-art computer vision methods is attributed to the development of deep neural networks. However it often comes at a computational price which may hinder their deployment. To alleviate this limitation, structured pruning is a well known technique which consists in removing channels, neurons or filters, and is commonly applied in order to produce more compact models. In most cases, the computations to remove are selected based on a relative importance criterion. At the same time, the need for explainable predictive models has risen tremendously and motivated the development of robust attribution methods that highlight the relative importance of pixels of an input image or feature map. In this work, we discuss the limitations of existing pruning heuristics, among which magnitude and gradient-based methods. We draw inspiration from attribution methods to design a novel integrated gradient pruning criterion, in which the relevance of each neuron is defined as the integral of the gradient variation on a path towards this neuron removal. Furthermore, we propose an entwined DNN pruning and fine-tuning flowchart to better preserve DNN accuracy while removing parameters. We show through extensive validation on several datasets, architectures as well as pruning scenarios that the proposed method, dubbed SInGE , significantly outperforms existing state-of-the-art DNN pruning methods.

## 1   Introduction

Deep neural networks (DNNs) are ubiquitous in modern solutions for most computer vision problems such as image classification [1], object detection [2] and semantic segmentation [3]. However, this performance was achieved at the price of high computational requirements and memory foot-print. As such, over-parameterization [4] is a common trait of well performing DNNs that may hinder their deployment on mobile and embedded devices. Furthermore, in the case of deployment on a cloud environment, latency and energy consumption are of paramount importance.

Consequently, compression and acceleration techniques aim at tackling the issue of DNN deployment. Among these methods, pruning approaches consist in removing individual weights (*unstructured* pruning) or entire computational blocks, such as neurons channels or filters (*structured* pruning) [5, 6, 7, 8]. The sparsity induced by pruning reduces both the computational cost and the memory foot-print of neural networks. To do so, there exists a wide variety of heuristics behind such pruning techniques. A few examples are: pruning at initialization [9], grid search [10, 11], magnitude-based [12] or redundancy based [7, 13] approaches. Among such heuristics, magnitude-based pruning remains the favoured one [14, 15, 16]. It consists in defining a metric to assess the relevance of each neuron in the network, with the goal to remove the least important ones while still preserving the predictive function as much as possible. An important limitation of these methods lies in the choice of this importance criterion: magnitude-based criteria [17] do not take into account the whole computations performed in the network (e.g. within the other layers) and gradient-based [18] criteria

36th Conference on Neural Information Processing Systems (NeurIPS 2022).

are intrinsically local within the neighborhood of a current value or set thereof: from this perspective, setting a value abruptly to zero might break this locality property.

To craft a better criterion, we borrow ideas from the field of DNN attribution [19]. These methods aim at understanding the behavior of a neural network, *i.e.*, in the case of a computer vision model, by providing visual cues of the most relevant regions in a image for the prediction of a network. Tools developed to explain individual predictions are also often called visual explanation techniques [20, 21, 22, 23, 24]. One example of such model is the Integrated Gradient method [24] that consists in defining the contribution of each input by the influence of marginal local changes in the input on the final prediction. This provides a fine-grained evaluation of the importance of each pixel of the image (alternatively, of an intermediate feature map) in the final decision.

Our work is based on the idea that DNN pruning and attribution methods share an important notion, namely that they both rely on the definition of an importance metric to compare several variables of a multidimensional prediction system: for pruning, to remove the least important DNN *parameters*, and, for attribution, to highlight the most important *pixels*. With this in mind, we propose to adapt the integrated gradient method for pruning purposes. Specifically, for each parameter (or set thereof, if we consider structured sparsity), we define its importance as an integral of the product between the norm of this weight and its attribution along a path between this weight value and a baseline (zero) value. By doing so, we avoid pathological cases which less sophisticated gradient-based methods are subjected to such as weights that can be reduced but not zeroed-out without harming the accuracy of the model. Furthermore, we embed the proposed integrated gradient method within a re-training framework to maximize the accuracy of the pruned DNN. We name our method SInGE , standing for **S**parsity *via* **In**tegrated **G**radients **E**stimation of neuron relevance. In short, the contributions of this paper are the following:

- We discuss the limitations of existing pruning heuristics, among which magnitude and gradient based methods. We draw inspiration from attribution methods to design an integrated gradient criterion for estimating the importance of each DNN weight.

- We entwine the updates of the importance measurement within the fine-tuning flowchart to preserve the better DNN accuracy while pruning.

- The proposed approach, dubbed SInGE, achieves superior accuracy *v.s.* pruning ratio on every tested dataset and architecture, compared with recent state-of-the-art approaches.

## 2 Related Work

### 2.1 Pruning

Pruning methods are often classified as either structured [25, 26, 27, 10, 28] (filters, channels or neurons are removed) or unstructured [15, 29, 30, 31] (single scalar weight values are set to zero). In practice, the former offers straightforward implementation for inference and immediate runtime benefits but at the price of a lower number of parameters removed. For instance, in GDP [32], weights are pruned with a learned gate that zeroes-out some channels for easier pruning post-training. In CCP [33], sparsity is achieved by evaluating the inter-channel dependency and the joint impact of pruned and preserved channels on the final loss function. In HAP [34], authors replace less sensitive channels based on the trace of the Hessian of predictive function with respect to the weights. Generally speaking, these methods rely on defining a criterion to estimate and compare the importance of weights in the networks, and remove the least important such candidates. A limitation of these methods is that the proposed criteria are usually only relevant within the neighborhood of the current value for a considered weight, which can be problematic since abruptly setting this weight value might violate this locality principle. In this work, we address this limitation by borrowing ideas from the DNN attribution field.

### 2.2 Attribution

Attribution methods, also referred to as visual explanation methods [20, 21, 22, 23, 24] measure the importance of each input feature on the prediction. Their use was motivated by the need for explainable models [19] as well as constrained learning [35]. We can classify attribution as either occlusion-based or gradient-based. The latter usually offers satisfactory results at a much lower

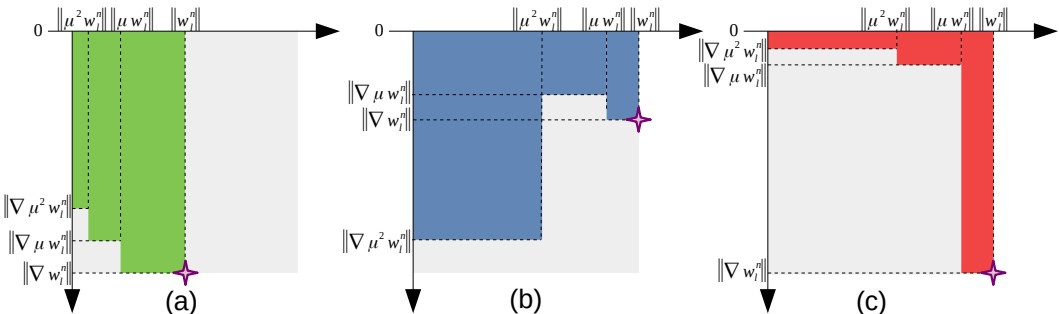

Figure 1: Illustration of possible limitations of traditional pruning criteria for 3 distinct cases and neurons (a,b,c). For a neuron $n$ at layer $l$ we plot the weights norm $||\mu^s w_l^n||$ and corresponding gradients norm $||\nabla \mu^s w_l^n||$ of different neurons (a, b and c) for different powers of $\mu^s \in\ ]0; 1[$ corresponding to a path towards zeroing out this neuron. Magnitude-based approaches **(a)** remove low magnitude neurons regardless of the sensitivity (gradient norm) of the predictive function w.r.t. these neurons. Gradient-based approaches **(b)** are limited by the intrinsic locality of the gradient, and abruptly setting a neuron weights to zero may break this locality principle. Conversely, our integrated gradient-based approach **(c)** will prune neuron although it initially has a high magnitude and gradient, integrating its gradient variations along a path down to zero magnitude.

computational cost. Considering that most DNNs for classification are derivable, Grad-CAM [36] computes the gradients of the predictive function with respect to feature maps and weights these gradients by the features. The resulting importance maps are then processed by a ReLU function to extract the features that should be increased in order to increase the value of the target class. Another gradient-based attribution of interest is Integrated-Gradients [24]. In this work, Sundararajan *et al.* propose to sum the gradients of the predictive function with respect to the feature maps over a uniform path in the feature space between feature at hand and a reference point. The resulting attribution maps are usually sharper than maps obtained by Grad-CAM. In the proposed method, we draw inspiration from these methods as we propose to integrate the (local) evolution of the pruning criteria throughout a path going from the current weight value down to a baseline (zero) value. This way, we can smoothly bring the most irrelevant weights down to zero even using intrinsically local criteria such as gradients or gradients per weight norm products.

## 3 Methodology

Let $F : \mathcal{D} \mapsto \mathbb{R}^{n_o}$ be a feed forward neural network defined over a domain $\mathcal{D} \subset \mathbb{R}^{n_i}$ (e.g. the training dataset in most instances) and an output space $\mathbb{R}^{n_o}$. The operation performed by a layer $f_l$, for $l \in \{1, \ldots, L\}$, is defined by the corresponding weight tensor $W_l \in \mathcal{A}^{n_{l-1} \times n_l}$ where $\mathcal{A}$ is simply $\mathbb{R}$ in the case of fully-connected layers and $\mathbb{R}^{k \times k}$ in the case of a $k \times k$ convolutional layer. For the sake of simplicity, we assume in what follows that $\mathcal{A} = \mathbb{R}$, *i.e.* we remove neurons as represented by their weight vectors.

### 3.1 Simple baseline pruning criteria

One major component of pruning methods lies in the definition of an importance measurement for each neuron. The most straightforward such criterion is based on the magnitude of the weight vectors. In such a case, the importance criterion $C_{L^p}$ based on the $L^p$ norm $\| \cdot \|_p$, is defined as:

$$C_{L^p} : (W_l, F, \mathcal{D}) \mapsto (\|W_l^n\|_p)_{n \in \{1, \ldots, n_l\}} \tag{1}$$

where $W_l^n$ is the $n^{\text{th}}$ column of $W_l$, corresponding to the weight values associated with the $n^{\text{th}}$ neuron of layer $f_l$. The transformation $C_{L^p}$ operates layer per layer and independently of the rest of network $F$ and the domain $\mathcal{D}$. Intuitively, $C_{L^p}$ assumes that the least important neurons are the smallest in norm because such neurons have a lower impact on the predictions of $F$. Such a simple criterion however face limitations: consider for instance the two first neurons (a) and (b) depicted in Figure 1 by the purple stars in two-dimensional spaces as function of their magnitude and gradient norm respectively denoted $\|W_l^n\|$ and $\|\nabla W_l^n\|$, for simplicity. However, we can clearly see how these local measurements provide a wrong evaluation of the cost of pruning these neurons. In such a case,

pruning according to $C_{L^p}$ will remove neuron (a) regardless of the fact that the predictive function $F$ will be very sensitive to small modification of this neuron, as indicated by the large value of its gradients. This is however not the case with the gradient-based pruning criterion $C_{\nabla^p}$ defined as:

$$C_{\nabla^p} : (W_l, F, \mathcal{D}) \mapsto \left( \left\| \nabla_{W_l^n} F(\mathcal{X} \in \mathcal{D}) \right\|_p \right)_{n \in \{1, \ldots, n_l\}} \tag{2}$$

where $\nabla_{W_l^n} F(\mathcal{X} \in \mathcal{D})$ is the gradient of $F$ with respect to $W_l$, evaluated on $\mathcal{X}$ a sample from $\mathcal{D}$. Intuitively, the latter measurement puts more emphasis on neurons that can be modified without directly altering the predictive function $F$. However, a neuron may have a low gradient norm and still strongly contribute to the predictive function, e.g. in the case where the weight is large as in the case of neuron (b) on Figure 1. To handle this, the norm $\times$ gradient criterion $C_{L^p \times \nabla^p}$ straightforwardly combines the best of both worlds:

$$C_{L^p \times \nabla^p} : (W_l, F, \mathcal{D}) \mapsto \left( \left\| W_l^n \right\|_p \times \left\| \nabla_{W_l^n} F(\mathcal{X} \in \mathcal{D}) \right\|_p \right)_{n \in \{1, \ldots, n_l\}} \tag{3}$$

### 3.2 Integrating gradients towards neuron removal

The importance criterion $C_{L^p \times \nabla^p}$ in Equation (3) faces another kind of limitation. due to the local nature of gradient information: if we consider neuron (b) on Figure 1, this neuron may initially (*i.e.* within a neighborhood of the purple star) have a low gradient norm or even low magnitude per gradient norm product. However, the gradient becomes larger as we bring this value down to 0. This is due to the fact that $\nabla_{W_l^n} F(\mathcal{X} \in \mathcal{D})$ only holds within a neighborhood of $W_l^n$ current value, and abruptly setting this neuron weights to zero may very well violate this locality principle. Thus, inspired from attribution methods, we propose a more global integrated gradient criterion. Formally, for neuron $n$ of a layer, $l$, we define $\mathcal{I}_l^n$ as the following integral:

$$\mathcal{I}_l^n = \int_{\mu=0}^{1} \left\| \nabla_{\mu W_l^n} F(\mathcal{X} \in \mathcal{D}) \right\|_p d\mu \tag{4}$$

Intuitively, we measure the cost of progressively decaying the weights of neuron $n$ and integrating the gradient norm throughout. In practice, we approximate $\mathcal{I}_l^n$ with the following Riemann integral:

$$C_{\mathrm{IG}^p} : (W_l, F, \mathcal{D}) \mapsto \left( \sum_{s=0}^{S} \left\| \mu^s W_l^n \right\|_p \times \left\| \nabla_{\mu^s W_l^n} F(\mathcal{X} \in \mathcal{D}) \right\|_p \right)_{n \in \{1, \ldots, n_l\}} \tag{5}$$

where $\mu \in ]0; 1[$ denotes an update rate parameter. $C_{\mathrm{IG}^p}$ approximates $(\mathcal{I}_l^n)_{n \in \{1, \ldots, n_l\}}$ up to a multiplicative constant. Practically, this criterion measures the cost (as expressed by its gradients) of progressively bringing $W_l^n$ down to 0 by $S$ successive multiplication with the update rate parameter $\mu$: the higher $\mu$, the more precise the integration at the expanse of increasing number of computations $S$. Also note that, similarly to Equation 2, we can get rid of the weight magnitude term in Equation 5 to obtain criterion $C_{\mathrm{SG}^p}$, based on the sum of gradient norms. Explicitly, we get $C_{\mathrm{SG}^p} : (W_l, F, \mathcal{D}) \mapsto$ $\left( \sum_{s=0}^{S} \left\| \nabla_{\mu^s W_l^n} F(\mathcal{X} \in \mathcal{D}) \right\|_p \right)_{n \in \{1, \ldots, n_l\}}$. In the case depicted on Figure 1, we will prune neuron (c) as its gradient quickly diminishes as its magnitude becomes lower, despite high initial values for both magnitude and gradient. Thus, the proposed integrated gradients criterion $C_{\mathrm{IG}^p}$ allows to take into account both the magnitude of a neuron's weights and the sensitivity of the predictive function w.r.t. small (local) variations of these weights. Furthermore, it measures the cost of removing this neuron by smoothly decaying it, re-estimating the gradient value at each step, hence preserving the local nature of gradients.

### 3.3 Entwining neuron pruning and fine-tuning

In order to preserve the accuracy of the network $F$, we alternate between removing the neurons and fine-tuning the pruned network using classical stochastic optimization updates. More specifically, given $\rho$ a global pruning target for the whole network $F$, we define layer-wise pruning objectives

$(\rho_l)_{l \in \{1,\ldots,L\}}$ such that $\sum_{l=1}^{L} \rho_l \times \Omega(W_l) = \rho \times \Omega(F)$ where $\Omega(W_l)$ and $\Omega(F)$ denote the number of parameters in $W_l$ and $F$, respectively. Similarly to [13], we tested several strategies for the per-layer pruning rates and kept their per-block strategy. Then, we sequentially prune each layer, starting from the first one, by first evaluating the relevance of each neuron $(C_{\mathrm{IG}^p}(W_l, F, \mathcal{D}))_{n \in \{1,\ldots,n_l\}}$ (with parameter $\mu$) in layer $l$. We then rank the neurons by ascending numbers of importance and select the first, least important one. Notice at this point that if we remove neuron $n$ we have to recompute the criterion $C_{\mathrm{IG}^p}$ for all other neurons: in fact, during the first pass, the gradients $\nabla_{\mu^s W_l^n} F(\mathcal{X} \in \mathcal{D})$ were computed with $W_l^n \neq 0$ and are bound to be altered with the removal of neuron $n$, thus affecting the order of the $n_l - 1$ remaining neuron importance. Last but not least, once layer $l$ is pruned, we perform $O$ finetuning steps (which corresponds to $O$ gradient descent optimization steps) to retain the network accuracy. This method, dubbed SInGE for **S**parsity *via* **In**tegrated **G**radients **E**stimation of neuron importance, is summarized in Algorithm 1.

---

**Algorithm 1** SInGE Algorithm

---

**Require:** neural network $F$, hyper-parameters : $O$, $\mu$ and $(\rho_l)_{l \in \{1,\ldots,L\}}$ and dataset $\mathcal{D}$
    **for** $l \in \{1, \ldots, L\}$ **do**
        **while** pruning_rate$(W_l) \leq \rho_l$ **do**                            $\triangleright$ wait until we reach the goal
            evaluate $M \leftarrow C_{\mathrm{IG}^p}(W_l, F, \mathcal{D})$                  $\triangleright$ magnitude estimation
            find $n = \arg\min\{M\}$                        $\triangleright$ find the neuron to prune
            set $W_l^n \leftarrow 0$                          $\triangleright$ the pruning is performed here
            **for** $o \in \{1, \ldots, O\}$ **do**
                finetune the whole network $F$ over a batch from $\mathcal{D}$
            **end for**
        **end while**
    **end for**

---

Empirically, as we show through a variety of experiments that the proposed integrated gradients-based neuron pruning, along with efficient entwined fine-tuning allows to achieve superior accuracy *vs.* pruning rate trade-offs, as compared to existing methods.

## 4 Experiments

First, we introduce our experimental setup, including the datasets and architectures as well as the implementation details to ensure reproducibility of the results. Second, we validate our approach on Cifar10 dataset by showing the interest of the proposed integrated gradient criterion, as well as the entwined pruning and fine-tuning scheme. We also compare our results with existing approaches on Cifar10. Last but not least, we demonstrate the superior performance of our SInGE method on several architectures on ImageNet compared with state-of-the-art approaches for both structured and unstructured pruning.

### 4.1 Experimental setup

**Datasets and Architectures:** we evaluate our models on the two *de facto* standard datasets for architecture compression, *i.e.* Cifar10 [37] and ImageNet [38]. We use the standard evaluation metrics for pruning, *i.e.* the $\%$ of removed parameters as well as the $\%$ of removed Floating-point operations (FLOPs). We apply our approach on ResNet 56 ([1] with 852K parameters and accuracies $93.46\%$) on Cifar10 and ResNet 50 ([1] with 25M parameters and 76.17 accuracy on ImageNet), as well as MobileNet v2 [39] backbone on ImageNet with 71.80 accuracy and 3.4M parameters.

**Implementation Details:** our implementations are based on tensorflow and numpy python libraries. We measured the different pruning criteria using random batches $\mathcal{X}$ of 64 training images for both Cifar10 and ImageNet and fine-tuned the pruned models with batches of size 128 and 64 for Cifar10 and ImageNet, respectively. The number of optimization steps varies from $1k$ to $5k$ on Cifar10 and from $5k$ to $50k$ on ImageNet, while the original models were trained with batches of size 128 and stochastic gradient descent of $78k$ and $2m$ steps on Cifar10 and ImageNet, respectively. All experiments were performed on NVidia V100 GPU. We evaluate our approach both for structured and unstructured pruning: for the former, we use $\mu = 0.9$ and $\mu = 0.95$ for ImageNet and Cifar10,

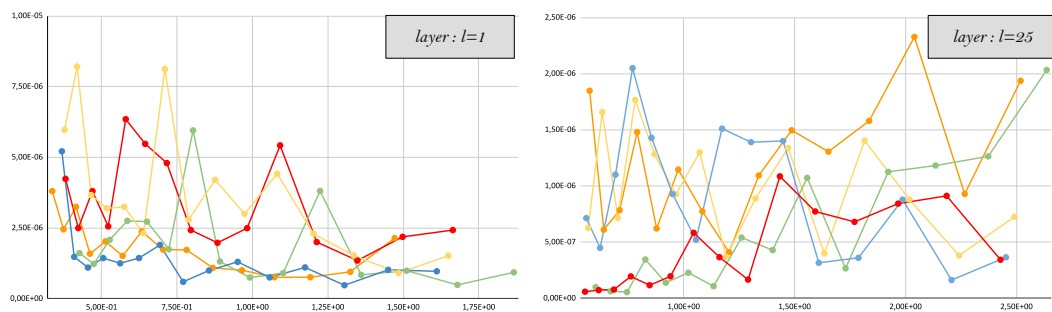

Figure 2: Visualization, for 5 random neurons and two different layers of a ResNet 56 trained on Cifar10, of the evolution of $\left\|\nabla_{\mu^s W_l^n} F(\mathcal{X} \in \mathcal{D})\right\|_p$ (y axis) as the magnitude $\|\mu^s W_l^n\|_p$ (x axis) is brought to 0.

Table 1: Pruning and accuracy performance of the different pruning criterion on a ResNet 56 trained on Cifar10. without fine-tuning. We also report the standard deviation over multiple runs.

| Pruning target (% FLOPS / parameters) | pruning criterion | top-1 accuracy |
|---|---|---|
| 0.0 / 0.0 | baseline | 93.46 |
| 73.03 / 75.00 | magnitude $C_{L^1}$ | $42.01 \pm 0.41$ |
| | magnitude $C_{L^2}$ | $42.35 \pm 0.38$ |
| | gradients $C_{\nabla^2}$ | $77.68 \pm 0.52$ |
| | magnitude $\times$ grad $C_{L^2 \times \nabla^2}$ | $92.36 \pm 0.17$ |
| | integrated gradients $C_{SG^2}$ | $93.01 \pm 0.07$ |
| | integrated magnitude $\times$ grad $C_{IG^2}$ | $\mathbf{93.23} \pm 0.23$ |
| 86.46 / 85.00 | magnitude $C_{L^1}$ | $19.14 \pm 0.82$ |
| | magnitude $C_{L^2}$ | $19.13 \pm 0.09$ |
| | gradients $C_{\nabla^2}$ | $28.31 \pm 1.75$ |
| | magnitude $\times$ grad $C_{L^2 \times \nabla^2}$ | $90.28 \pm 0.18$ |
| | integrated gradients $C_{SG^2}$ | $91.90 \pm 0.15$ |
| | integrated magnitude $\times$ grad $C_{IG^2}$ | $\mathbf{92.80} \pm 0.30$ |
| 88.10 / 90.00 | magnitude $C_{L^1}$ | $10.00 \pm 1<$ |
| | magnitude $C_{L^2}$ | $10.00 \pm 1<$ |
| | gradients $C_{\nabla^2}$ | $10.00 \pm 1<$ |
| | magnitude $\times$ grad $C_{L^2 \times \nabla^2}$ | $10.00 \pm 1<$ |
| | integrated gradients $C_{SG^2}$ | $75.38 \pm 1.28$ |
| | integrated magnitude $\times$ grad $C_{IG^2}$ | $\mathbf{84.54} \pm 0.91$ |

respectively. For unstructured pruning, we use $\mu = 0.8$ for ImageNet. In all experiments we performed batch-normalization folding from [40] and measured the pruning ratio using the same metric as SOSP [41].

## 4.2 Empirical Validation

**Pruning Criterion Validation:** In Figure 2, we illustrate the evolution of $\left\|\nabla_{\mu^s W_l^n} F(\mathcal{X} \in \mathcal{D})\right\|_p$ (y axis) as the magnitude $\|\mu^s W_l^n\|_p$ (x axis) is brought to 0. This observation confirms the limitations of gradient-based criteria pinpointed in Section 3.2: as the neuron magnitude is progressively decayed, the gradient norm (e.g. yellow curves on both plots, as well as red on the left and blue one on the right plot) for these neurons rise significantly, making these neurons bad choices for removal despite low initial gradient values. This empirical observation suggests that intuition behind the proposed criterion $C_{IG^p}$ is valid. Table 1 draws a comparison between the different criteria introduced in Section 3, applied to prune a ResNet 56 on Cifar10. More specifically, given a percentage of removed operations (or equivalently, percentage of removed parameters), we compare the resulting accuracy

Table 2: Comparison between post-pruning and entwined pruning and fine-tuning on a ResNet 56 on Cifar10.

| % Pruning target (% FLOPS / parameters) | fine-tuning | # steps | top-1 accuracy |
|---|---|---|---|
| | post-pruning | 1000 | 92.59 |
| | entwined | 1000 | **93.18** |
| 86.46 / 85.00 | post-pruning | 2000 | 92.66 |
| | entwined | 2000 | **93.25** |
| | post-pruning | 5000 | 93.13 |
| | entwined | 5000 | **93.31** |
| | post-pruning | 1000 | 77.2 |
| | entwined | 1000 | **85.38** |
| 88.10 / 90.00 | post-pruning | 2000 | 80.89 |
| | entwined | 2000 | **87.52** |
| | post-pruning | 5000 | 86.39 |
| | entwined | 5000 | **90.02** |

without fine-tuning. We observe similar trends for the 3 pruning targets: First, using euclidean norm performs slightly better than $L_1$: thus, we set $p = 2$ in what follows. Second, using gradient instead of magnitude-based criterion allows to significantly improve the accuracy given a pruning target. Third, the magnitude × gradient criterion $C_{L^2 \times \nabla^2}$ allows to better preserve the accuracy by combining the best of both worlds: for instance, with $85\%$ parameters removed, applying $C_{L^2 \times \nabla^2}$ increases the accuracy by $51.97$ points compared with $C_{\nabla^2}$. However, those simple criteria face limitations particularly in the high pruning rate regime ($90\%$ parameters removed), where the accuracy of the pruned network falls to chance level. Conversely, the proposed integrated gradient-based criteria $C_{SG^2}$ and, *a fortiori* $C_{IG^2}$ allows to preserve high accuracies in such a case. Overall, $C_{IG^2}$ is the best criterion, allowing to preserve near full accuracy with both $85\%$ and $85\%$ removed parameters, and $85.38\%$ accuracy with $90\%$ removed parameters, outperforming the second best method by $8.18$ points. For this reason, we will use this criterion ($C_{IG^2}$) in the following experiments. As the pruning rate increases, the cost of removing a neuron increases and any ill-advised selection of neuron to remove has a growing impact on the accuracy. Consequently, the standard deviation increases as the pruning rate increases this is due to the network expressivity going down.

**Fine-tuning Protocol Validation:** Table 2 validates our entwined pruning and fine-tuning approach with different pruning targets and number of fine-tuning steps. Specifically, for a given total number of fine-tuning step, we either perform all these steps at once *post-pruning* or alternatively spread them evenly after pruning each layer in an entwined fashion, as described in Section 3.3. First, we observe that simply increasing the number of fine-tuning steps vastly improves the accuracy of the pruned model, particularly in the high % removed parameters regime. Moreover, entwining pruning and fine-tuning performs consistently better than fine tuning after pruning. This suggests that recovering the accuracy is an easier task when performed frequently over small modifications rather than once over a significant modification.

**Comparison with state-of-the-art approaches on Cifar10:** our approach relies on removing the least important neurons, as indicated by criterion $C_{IG^2}$. We compare with similar recent approaches such as LP [42] and DPF [29] as well as other heuristics such as training neural networks in order to separate neurons for easier pruning post-training (HAP [34], GDP [32]) or similarity removal (RED [13] or LDI [31]). We report the results in Table 3 for two accuracy set-ups: lossless pruning (accuracy identical to the baseline model) and lossy pruning ($\approx 2$ points of accuracy drop). The proposed SInGE method significantly outperforms other existing methods by achieving $1.3\%$ higher pruning rate in the lossy setup and a considerable $8.1\%$ improvement in lossless pruning rate. As such, it bridges the gap with unstructured methods such as LP [42] and DPF [29]. This demonstrates the quality of the proposed method.

Table 3: State-of-the-art pruning methods performance on ResNet 56 on Cifar10.

| top1 accuracy | pruning method | structured | % parameters removed |
|---|---|---|---|
| 91.5 $\pm$ 0.1 | RED [13] | ✓ | 85.0 |
| | LP [42] | ✓ | 84.0 |
| | LP [42] | ✗ | 92.6 |
| | LDI [31] | ✓ | 88 |
| | DPF [29] | ✗ | 90.0 |
| | HAP [34] | ✓ | 90.0 |
| | SInGE (ours) | ✓ | **91.3** $\pm$ 0.27 |
| 93.5 $\pm$ 0.1 | GDP [32] | ✓ | 65.6 |
| | HAP [34] | ✓ | 76.2 |
| | SInGE (ours) | ✓ | **84.3** $\pm$ 0.71 |

Table 4: Comparison between existing structured pruning performance on ResNet 50 on ImageNet. In both the low ($< 50\%$ parameters removed) and high ($> 50\%$) pruning regimes, SInGE achieves remarkable results.

| Method | % params rm | % FLOPS rm | accuracy |
|---|---|---|---|
| baseline | 0.00 | 0.00 | 76.15 |
| Hrank (CVPR 2020) [48] | 36.67 | 43.77 | 74.98 |
| RED (NeurIPS 2021) [13] | 39.6 | 42.7 | 76.1 |
| HAP (WACV 2022) [34] | 44.59 | 33.82 | 75.12 |
| SRR-GR (CVPR 2021) [28] | - | 45 | 75.76 |
| SOSP (ICLR 2021) [41] | 49 | 45 | 75.21 |
| SRR-GR (CVPR 2021) [28] | - | 55 | 75.11 |
| SInGE | **50.80** $\pm$ 0.02 | **57.35** $\pm$ 0.11 | **76.05** $\pm$ 0.07 |
| RED (NeurIPS 2021) [13] | 54.7 | 55.0 | 71.1 |
| SOSP (ICLR 2021) [41] | 54 | 51 | 74.4 |
| GDP (ICCV 2021) [32] | - | 55 | 73.6 |
| HAP (WACV 2022) [34] | **65.26** | 59.56 | 74.0 |
| OTO (NeurIPS 2021) [43] | 64.1 | 65.2 | 73.3 |
| GFP (ICML 2021) [49] | - | 65.0 | 73.94 |
| SInGE | 63.78 $\pm$ 0.01 | **65.96** $\pm$ 0.21 | **74.7** $\pm$ 0.31 |

## 4.3 Performance on ImageNet

**Structured Pruning:** Table 4 summarizes results obtained by current state-of-the-art approaches in structured pruning. For clarity we divided these results in the low ($<50\%$ parameters removed, where the methods are often lossless) and high pruning regime ($>50\%$ parameters removed with significant accuracy loss). In the low pruning regime, the proposed SInGE method manages to remove slightly more than $50\%$ parameters ($57.35\%$ FLOPS) with nearly no accuracy loss, which significantly improves over existing approaches. Second, in the high pruning regime, other methods such as OTO [43] and HAP [34] recently improved the pruning rates by more than 10 points over other techniques such as GDP [32] and SOSP [41]. Nonetheless, SInGE is competitive with these methods and achieve a higher FLOP reduction while maintaining a higher accuracy.

We also evaluated the proposed method on the more compact (thus generally harder to prune) MobileNet V2 architecture. Results and comparison with existing approaches are shown in Table 5. We consider three pruning goals of $\approx 30\%$, $\approx 40\%$ and $\approx 50\%$ parameters removed. First, with near lossless pruning, we achieve results that are comparable to ManiDP-A [44] and Adapt-DCP [45] with a marginal improvement in accuracy. Second, when targeting 40% parameters removed we improve by $0.89\%$ the accuracy with 2.25% less parameters removed as compared to MDP [46]. Finally, in the higher pruning rates, we improve by $0.25\%$ the accuracy with marginally more parameters pruned than Accs [47].

Table 5: Comparison with existing structured pruning methods on MobileNet V2 backbone for ImageNet.

| goal | Method | % params rm | % FLOPS rm | accuracy |
|---|---|---|---|---|
| - | baseline | 0.00 | 0.00 | 71.80 |
| 30% | CBS (arxiv 2022) [50] | 30.00 | - | 71.48 |
| | Adapt-DCP (TPAMI 2021) [45] | **35.01** | 30.67 | 71.4 |
| | ManiDP-A (CVPR 2021) [44] | - | **37.2** | 71.6 |
| | SInGE | 30.96 | 31.54 | **71.67** $\pm$ 0.06 |
| 40% | CBS (arxiv 2022) [50] | 40.00 | - | 69.37 |
| | MDP (CVPR 2020) [46] | **43.15** | - | 69.58 |
| | SInGE | 40.90 | 42.30 | **70.47** $\pm$ 0.09 |
| 50% | CBS (arxiv 2022) [50] | 50.00 | - | 62.96 |
| | Adapt-DCP (TPAMI 2021) | - | 45.0 | 64.13 |
| | ManiDP-A (CVPR 2021) | - | 48.8 | 69.62 |
| | Accs (arxiv 2021) [47] | 50.00 | - | 69.76 |
| | GFP (ICML 2021) [49] | - | 50.0 | 69.16 |
| | SInGE | **50.13** | **48.90** | **70.01** $\pm$ 0.22 |

Table 6: Comparison with existing unstructured pruning techniques on ResNet 50 on ImageNet.

| Method | % params rm | % FLOPS rm | top1 accuracy |
|---|---|---|---|
| DS (NeurIPS 2021) [51] | 80.47 | 72.13 | 76.15 |
| GMP (arxiv 2019) [52] | 80.08 | - | 76.15 |
| STR (ICML 2020) [53] | 79.69 | 81.17 | 76.00 |
| RigL (ICML 2020) [54] | 80.08 | 58.92 | 75.00 |
| SInGE | 80.00 | 82.21 | 75.12 |
| SInGE | 90.00 | 86.96 | 73.77 |

**Unstructured Pruning:** While being harder to leverage, unstructured pruning usually enables significantly higher pruning rates. Table 6 lists several state-of-the-art pruning methods evaluated on ResNet 50. We observe a common threshold in performance around 80% parameters and FLOPs removed among state-of-the-art techniques. However, the proposed SInGE method manages to achieve very good accuracy of 73.77% while breaking the barrier of pruning performance at 90% parameters removed and almost 87% FLOPs removed. These results in addition to the previous excellent results obtained on structured pruning confirm the versatility of the proposed criterion and method for both structured and unstructured pruning.

# 5 Conclusion

In this paper, we pinpointed some limitations of some classical pruning criteria for assessing neuron importance prior to removing them. In particular, we showed that magnitude-based approaches did not consider the sensitivity of the predictive function w.r.t. this neuron weights, and that gradient-based approaches were limited to the locality of the measurements. We drew inspiration on recent DNN attribution techniques to design a novel integrated gradients criterion, that consists in measuring the integral of the gradient variation on a path towards removing each individual neuron. Furthermore, we proposed to entwine this criterion within the fine-tuning steps. We showed through extensive validation that the proposed method, dubbed SInGE, achieved superior accuracy *v.s.* pruning ratio as compared with existing approaches on a variety of benchmarks, including several datasets, architectures, and pruning scenarios.

Future work will involve introducing stochasticity in the model weights, similarly to [55], in order to smooth the decision function and ultimately the neuron relevance criterion. Lastly, we will combine our approach with existing similarity-based pruning methods as well as with other DNN acceleration techniques, e.g. tensor decomposition or quantization techniques.

## Acknowledgments

This work has been supported by the french National Association for Research and Technology (ANRT), the company Datakalab (CIFRE convention C20/1396) and by the French National Agency (ANR) (FacIL, project ANR-17-CE33-0002). This work was granted access to the HPC resources of IDRIS under the allocation 2022-AD011013384 made by GENCI.

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
