# OpenReview forum: "SInGE: Sparsity via Integrated Gradients Estimation of Neuron Relevance"
_NeurIPS.cc/2022/Conference — NeurIPS 2022 Accept_

### Official Review · Reviewer_BMyz · 2022-06-29

**Rating:** 6
**Confidence:** 4
**Soundness:** 3 good
**Presentation:** 3 good
**Contribution:** 3 good

**Summary:**

This paper proposes a novel pruning method based on an integrated gradient pruning criterion, which combines a magnitude and gradient based criterion and integrates the product over the path of the neuron removal.

In addition, the work proposes a fine-tuning flowchart to improve the performance of the fine-tuned network.  This strategy is an iterative layer-wise approach which sets the pruning ratios for each block separately.

Lastly, a comparison of SInGE on ResNet-56 on Cifar10 and ResNet-50 and MobileNet-V2 on ImageNet shows state-of-the-art performance.


**Questions:**

I quite like the idea of the paper and would be willing to increase my score if the authors can address my concerns. In addition to the concerns mentioned above, I have some more minor concerns:

Why do you not compare to the same methods on all datasets? It seems that some of the methods compared to on ImageNet also provide results for CIFAR10. Further, some recent structured pruning works such as Group Fisher Pruning [A] were not considered for comparison

Why did you choose to use the product of the norms of the gradient of the output of the NN and the weights? One could imagine other gradients i.e., the gradient of the loss.


[A] Liu, Liyang, et al. "Group fisher pruning for practical network compression." International Conference on Machine Learning. PMLR, 2021.


**Limitations:**

Yes

**Strengths And Weaknesses:**

Strengths:

* I like the idea of an integration criterion and according to Table 1 it seams that this is the main driver for the superior performance of SInGE. To the best of my knowledge this is the first method to use an integration criterion for (structured) pruning.

* The Integration criterion is well motivated, and the ablations support the design choices made by the authors.

* SInGE achieves state-of-the-art performance on CIFAR10 and ImageNet

Weaknesses:

* The main weakness that I see is that the work does not explain how the FLOPS and parameters of the pruned network are calculated. This is a major issue because the work does not directly compare to other state-of-the-art methods but merely quotes the numbers. Since the calculation of the FLOPS and parameters of a pruned network is non-trivial (see Appendix D of SOSP [43]) This can have a huge impact on the performance and could reduce the performance significantly.

* The second main weakness is that the work applies an iterative fine-tuning scheme but compares to single-shot pruning methods such as HAP, SOSP, OTO and SRR-GR. Since the paper shows in Table 2 that an iterative pruning scheme seems to improve the overall performance significantly, this would give SInGE an advantage and makes a direct comparison of the pruning criterions harder.

* Further, the authors do not mention the computational effort to evaluate the criterion. Due to the Riemannian Integral this should be significantly higher compared to other gradient, magnitude or first-order based methods.

* Lastly a direct comparison to at least one competing method within the same pruning pipeline would provide the possibility to better disentangle the performance gains from the fine-tuning pipeline and criterion.

---

> ### Author Response · Authors · 2022-08-02
> **Answer to Reviewer BMyz (1/2)**
>
> **The main weakness that I see is that the work does not explain how the FLOPS and parameters of the pruned network are calculated. This is amajor issue because the work does not directly compare to other state-of-the-art methods but merely quotes the numbers. Since the calculation of the FLOPS and parameters of a pruned network is non-trivial (see Appendix D of SOSP \[43\]) This can have a huge impact on the performance and could reduce the performance significantly.**
>
> In their appendix, the authors of SOSP explain how some pruning methods measure the pruning ratio and MACs count in two different manners. They have an extended discussion on a problem that is well known in the pruning community \"how do we prune skip connections\". To put it simply: if two layers were to be added but pruned independently (which is almost systematically the case) then if the channel pruned do not match, one can't actually remove the channels as they would still be required for the rest of the network. More specifically, for example, let's consider two layers $f_1$ and $f_2$ that are added in a skip connection and we want to prune neuron $1$ from $f_1$ and neuron $2$ from $f_2$ then their output would still be:
>
> $f_1(x) + f_2(x) = \begin{pmatrix}0 &+& {(f_2(x))}_1 \\\\ {(f_1(x))}_2 &+& 0 \\\\ {(f_1(x))}_3 &+& {(f_2(x))}_3 \\\\ \vdots & & \vdots \end{pmatrix}$
>
> In such a case, no channel is effectively pruned. For this reason, in SOSP \[43\], the authors argue that one should not count the pruning in such instances as it can't be applied in practice. For information, OTO \[42\] actually addresses this problem by forcing the simultaneous pruning of channels that should be matched. In our own study we apply the same pruning ratio measurement at SOSP as we think it is the only valid one plus it is more difficult than the approximate one.
>
> We added this point for the sake of clarity in the revised version of our work.
>
> **The second main weakness is that the work applies an iterative fine-tuning scheme but compares to single-shot pruning methods such as HAP, SOSP, OTO and SRR-GR. Since the paper shows in Table 2 that an iterative pruning scheme seems to improve the overall performance significantly, this would give SInGE an advantage and makes a direct comparison of the pruning criterions harder.**
>
> Indeed, in our evaluations we apply the proposed entwined importance criterion estimation and fine-tuning scheme, which improves the accuracy, as shown in Table 2. However, please bear in mind that our method requires so much fewer fine-tuning steps as compared to other methods, e.g. HAP, SOSP, OTO or SRR-GR. These methods involve more than 1 million optimization step with batches of 256 elements or several million steps for smaller batch-sizes. On the other hand (as stated in our implementation details section), we benchmarked SInGE with fewer (50k) optimization steps. Considering the huge discrepancy in terms of number of optimisation steps, we believe that the comparison is not only fair to state-of-the-art methods but shows the efficiency of the proposed approach.
>
> **Further, the authors do not mention the computational effort to evaluate the criterion. Due to the Riemannian Integral this should be significantly higher compared to other gradient, magnitude or first-order based methods.**
>
> Indeed the proposed integrated gradient criterion comes at the expanse of an additional (linearly increasing with the number of integration steps) computational cost. Please note that other state-of-the-art methods do not report their importance measurement time as they often require second order derivations or even very large batches for gradient estimation, also involving additional computational overhead.

---

> > ### Author Response · Authors · 2022-08-02
> > **Answer to Reviewer BMyz (2/2)**
> >
> > Furthermore, to specifically anwser your concern, we propose a novel benchmark on Cifar10 where we work at a constant processing (pruning+fine-tuning) time: specifically, as we empirically measure that 1 integrated gradient step costs approximately half the time of a fine-tuning step, we modulate the numbers of fine-tuning steps to reach a similar processing time, and compare the accuracy of the pruned networks. Results can be found below:
> >
> > |                        method                        | pruning rate / FLOPs removed |          accuracy         |
> > |:----------------------------------------------------:|:----------------------------:|:-------------------------:|
> > |                  magnitude $C_{L^1}$                 |         86.46 / 85.00        |      69.42 $\pm$ 2.11     |
> > |                  magnitude $C_{L^2}$                 |         86.46 / 85.00        |      75.38 $\pm$ 2.76     |
> > |               gradients $C_{\nabla^2}$               |         86.46 / 85.00        |      79.55 $\pm$ 1.97     |
> > |    magnitude $\times$ grad $C_{L^2\times\nabla^2}$   |         86.46 / 85.00        |      91.05 $\pm$ 0.03     |
> > |        integrated gradients $C_{\text{SG}^2}$        |         86.46 / 85.00        |      92.01 $\pm$ 0.13     |
> > | integrated magnitude $\times$ grad $C_{\text{IG}^2}$ |         86.46 / 85.00        | **93.07** $\pm$ 0.06      |
> >
> > [Table 5: Comparison of the performance of the different pruning criterion on a ResNet 56 for Cifar10 under the same time constraint.]
> >
> > **Lastly a direct comparison to at least one competing method within the same pruning pipeline would provide the possibility to better disentangle the performance gains from the fine-tuning pipeline and criterion.**
> >
> > In Table 1 of the paper we compare several importance pruning criteria, showing the interest of the proposed integrated magnitude $\times$ gradients-based criterion, without any fine-tuning. Furthemore, given the best criterion, Table 2 highlight the interest of entwined importance estimation and fine-tuning compared with a traditional approach. We believe that these comparisons allow to disentangle the gains coming from either the criterion and the proposed entwined fine-tuning and importance measurement scheme.
> >
> > **Why do you not compare to the same methods on all datasets? It seems that some of the methods compared to on ImageNet also provide results for CIFAR10. Further, some recent structured pruning works such as Group Fisher Pruning \[A\] were not considered for comparison. \[A\] Liu, Liyang, et al. \"Group fisher pruning for practical network compression.\" International Conference on Machine Learning. PMLR, 2021.**
> >
> > We added Group Fisher Pruning (GFP) to the paper in Table 4-5. Note that despite the fact that their baseline ResNet-50 has a higher accuracy than ours, our proposed approach is still competitive, further highlighting the quality of the proposed results.
> >
> >
> > **Why did you choose to use the product of the norms of the gradient of the output of the NN and the weights? One could imagine other gradients i.e., the gradient of the loss.**
> >
> > Akin to attribution methods \[21,22,23,24,36\] we use the gradient of the network output w.r.t. each neuron to measure the sensitivity of the function w.r.t. this neuron. Our intuition is that while derivating the loss allows to measure wether a neuron contributes to correct classification, derivating the whole outputs also allows to considerate the classification uncertainty.

---

> > > ### Comment · Reviewer_BMyz · 2022-08-09
> > > **Thank you for the detailed response**
> > >
> > > Thank you for your detailed response. Some of my concerns were addressed.  Adding a more detailed description of the calculation of the FLOP count to the paper would increase clarity. Furthermore, additional comparisons to Group Fisher Pruning were added. Therefore, I will raise my score even though I am not fully convinced by the arguments provided by the authors for comparing SInGE to single-shot methods. I know that a completely fair comparison of different pruning methods can be challenging but I feel that a direct comparison of iterative and single-shot methods (despite similar training time) could favour the iterative procedure.

---

> > > > ### Author Response · Authors · 2022-08-09
> > > > **Thank you for your comment**
> > > >
> > > > Thanks for your comments and time spent reviewing the paper and rebuttal. We agree that the proposed entwined importance criterion estimation and fine-tuning provides a boost to SInGE accuracy ; however, please bear in mind that both steps (i.e. importance estimation via integrated gradients method and entwining pruning and fine-tuning steps) are contributions of our work, both being ingredients to SInGE performance. Also, to specifically answer your concern, we did our best our to disentangle (see Table 2 in the revised paper and Table 5 in our latest response to the rebuttal) the performance gains brought by both these steps.

---

### Official Review · Reviewer_kWFQ · 2022-07-01

**Rating:** 5
**Confidence:** 3
**Soundness:** 3 good
**Presentation:** 2 fair
**Contribution:** 3 good

**Summary:**

This paper proposes a new neuron pruning  method based on the Integrated Gradients (IG) attribution algorithm. The paper claims that IG-based pruning is more optimal than magnitude and local gradient-based methods since it accounts for the gradient effects globally along the path from 0-valued baseline weight to given input weight in each layer l. The paper performs impressive number of experiments for a number of structured and unstructured pruning techniques from the state of the art literature and shows that IG-based pruning outperforms those methods based on two evaluation metrics (% removed parameters and % removed floating point operators).
In addition to that the paper also proposes entwined pruning and fine-tuning procedure and shows that it help to obtain better accuracy while performing the pruning.
The experiments in the paper are conducted for ImageNet and Mobilenet V2 backbone for ImageNet and  CIFAR10 dataset.


**Questions:**


+ From Table 1 it is unclear what the authors mean by integrated gradients C_{SG}^2 and integrated magnitude x grad C_{IG}^2. It would be good if they could elaborate it further and also describe which of those two they used for the results in tables 3, 4, 5 and 6.
+ I wonder if Noisegrad paper which introduces stochasticity to model weights could also be relevant to this paper.
https://arxiv.org/pdf/2106.10185.pdf


**Limitations:**


I do not see any potential negative societal impact in this work

**Strengths And Weaknesses:**

**Strengths**

+ Overall the idea of the paper is clearly described, related work is properly cited, experimental setup and results are properly discussed
+ The paper shows strong empirical evidence that IG-based pruning outperforms magnitude, gradient-based approaches, as well as a number of other state of the art approaches from the neuron pruning literature.
+ The paper also shows that entwined fine-tuning helps to improve model accuracy regardless of pruning technique using over 80% pruning target.

**Weaknesses**

+ I think that in general the readability (fluency) of the paper can be further improved. Specifically, I felt that in the notation and description of the approaches, there is some level of expectations that the readers know the notations of structured pruning techniques and integrated gradients. For example, neither Figure 1 nor the context talking about Figure 1 describes what $\mu$ and $\mu^2$ are.
+ Figure 1 and its descriptions are a bit confusing. Originally, I was thinking that a, b and c are weight matrices but later the paper refers to a and b as two different weights. It would be good to improve the description of Figure 1.
+ Given that entwined fine-tuning is one of the important contributions of the paper it would be good to describe it in detail. It would be good to make it clear what fine-tuning over O steps mean and what we mean by fine-tuning the network F over a batch from D. It seems that the authors make an assumption that the readers are familiar with the specific fine-tuning approach that they are describing.
At this point from the given description it is hard to tell the amount of novelty in the entwined fine-tuning.
+ Table 1: It would be great if the authors explain relatively large std in integrated magnitude x grad C_{IG}^2 method

**Minor Comments**

+ On page 9 a notation, $N_n$ but I don’t see a description of notation $N$.
+ Table 6 has 2 SInGE rows but it is not clear what are the differences between those two SInGEs

---

> ### Author Response · Authors · 2022-08-02
> **Answer to Reviewer kWFQ**
>
> **I think that in general the readability (fluency) of the paper can be further improved. [...]  For example, neither Figure 1 nor the context talking about Figure 1 describes what $\mu$ and $\mu^2$ are.**
>
> To answer your concern and make the paper more standalone and understandable, we updated the caption of Figure 1 in the revised version and provided detailed explanations in the main body.
>
> **Figure 1 and its descriptions are a bit confusing. Originally, I was thinking that a, b and c are weight matrices but later the paper refers to a and b as two different weights. It would be good to improve the description of Figure 1.**
>
> As previously stated, we agree with the reviewer and improved the writing of the paper to this regard.
>
> **Given that entwined fine-tuning is one of the important contributions of the paper it would be good to describe it in detail. [...] It seems that the authors make an assumption that the readers are familiar with the specific fine-tuning approach that they are describing. At this point from the given description it is hard to tell the amount of novelty in the entwined fine-tuning.**
>
> In our work, we advocate for an entwined pruning (*via* importance measurement as prescripted by the proposed integrated gradient criterion) and fine-tuning scheme, that allows to more smoothly remove the least relevant weights and adapt the network by retaining its accuracy. The proposed entwined pruning and fine-tuning is simple: specifically, fine-tuning over $O$ steps simply corresponds to running $O$ steps of stochastic gradient descent (or any other optimizer) like in standard DNN training. Similarly, using a batch from $\mathcal{D}$ simply refers to a random batch from the training set or more generally from the domain $\mathcal{D}$. Nevertheless, we show empirically (see Table 2 in the paper) that it increases the performance in all tested configurations in practice.
>
> **Table 1: It would be great if the authors explain relatively large std in integrated magnitude x $grad C_{IG}^2$ method**
>
> Generally speaking, we observe that the standard deviation increases as the pruning ratio increases regardless of the pruning criterion (e.g. the std is also large with the gradient pruning criterion at $85.00\%$ as well as the integrated gradient criterion at $90.00\%$ parameters removed): indeed, in such challenging settings, the cost of removing a neuron increases and any ill-advised selection of neuron is going to have growing impact on the accuracy.
>
> **On page 9 a notation, $N_n$ but I don't see a description of notation $N$.**
>
> This is a mistake from our side. The notation referred by the reviewer should be $W_l^n$ as it corresponds to pruning the n$^{th}$ neuron of layer $l$. We corrected this element in the revised article.
>
> **Table 6 has 2 SInGE rows but it is not clear what are the differences between those two SInGEs**
>
> Thank you for pointing this out, we corrected this typo in the updated manuscript.
>
> **From Table 1 it is unclear what the authors mean by integrated gradients $C_{SG}^2$ and integrated magnitude x $grad C_{IG}^2$. It would be good if they could elaborate it further and also describe which of those two they used for the results in tables 3, 4, 5 and 6.**
>
> As stated line 142 in the original paper, the $C_{SG^p}$ can be derived from equation (5) by removing the weight magnitude term. Explicitly this leads to the following definition
>
> $C_{\text{SG}^p} :(W_l,F,\mathcal{D})\mapsto(\sum_{s=0}^S\|\nabla_{\mu^sW_l^n}F(\mathcal{X}\in\mathcal{D})|\_p)_{n\in\{1,\dots,n_l\}}$
>
> We add this explicit definition to the main body. Furthermore, as stated line 211-214, the $C_{IG^p}$ is the most efficient criterion according to our empirical validation. We clarify that we use the most efficient criterion in the other experiments.
>
> **I wonder if Noisegrad paper which introduces stochasticity to model weights could also be relevant to this paper. https://arxiv.org/pdf/2106.10185.pdf**
>
> This work could help us improve the current importance metric. In the article suggested by the reviewer, the study focuses on SmoothGrad which is a technique for explainable AI which is a broader topic than attribution from which we draw inspiration. The method consist in adding noise to the values evaluated with the explanation technique. More formally, in our study, this would correspond to the addition of a white nose to the weight values for both neuron norms and corresponding gradient norms. According to the cited paper: \"\[\...\] hypothesize that SmoothGrad perturbs the test sample in order to get a signal from the steepest part of the decision boundary.\". This could be intuitively linked to the measure of importance, in other words, the steepest part of the decision boundary is given by the most important computations. Thank you for pointing out this interesting paper, we added it to our future work section in the conclusion.

---

> > ### Comment · Reviewer_kWFQ · 2022-08-10
> > **Response to Authors**
> >
> > Thank you very much for addressing the questions. I think that perhaps it would be good to simplify the notations $C_{{SG}^P}$ and $C_{{IG}^P}$ . Also in the results you might also want to mentioned how reliable integral approximation is based on the completeness axiom.
> > Could this work be potentially extended to non-zero baselines as well and could we have a more generic formulation for that ?

---

> > > ### Author Response · Authors · 2022-08-10
> > > **Thank you for your reading and comment**
> > >
> > > Given the short amount of time before the end of the discussion period, we will update the manuscript with the suggested changes, that will allow to enhance the quality of the paper. As for using non-zero baselines, since the goal of the proposed work is the removal of neurons, in its current form we have to specifically consider zero values. However, it could be an interesting future work to use an arbitrary value (the same for all neurons of a layer), and remove them using redundancy-based pruning methods, e.g. RED [13]. Thank you for the suggestion and comments, we hope that this answer your concerns and further convince you on the benefits of the proposed work for the community.

---

### Official Review · Reviewer_kiMH · 2022-07-11

**Rating:** 6
**Confidence:** 2
**Soundness:** 3 good
**Presentation:** 2 fair
**Contribution:** 3 good

**Summary:**

Inspired by the integrated gradient method in attribution methods, the authors propose a new importance metric for pruning: an integral of the product between the norm of the parameter weight and its attribution along a path between this weight value and a baseline (zero) value. This metric captures more global view than previous magnitude-based and gradient-based pruning methods. Then, the authors devise the so-called SinGE method to intertwine the pruning with the network fine-tuning. Empirical studies are conducted to compare SinGE with both structured and unstructured pruning methods on Cifar 10 and ImageNet.

**Questions:**

1. As far as I undertand, the integration evaluation will take much time, so do the authors think that adding an experiment of the time efficiency necessary?
2. There are a few typos, for example, at the range of $\mu$ at line 138, the spellings of trade-offs at line 167, and the incompleteness of parentheses at line 181.
3. As I suppose the bolded numbers in the tables stand for the best results, some of those in Table 3 and 4 are not correctly labeled, which is a bit misleading.
4. The top-1 accuracy of integrated magnitude method without fine-tuning and with 1000-step entwined fine-tuning seems to be the same (both reported as 85.38). I’m wondering why there is no increase in the accuracy.
5. May I ask what does the notation “1<” stand for in Table 1?
6. As the performance of SinGE is very close to its counterpart in many cases, could the authors please provide the error bars for the results reported in Table 2-6?
7. May I ask whether the results in Table 3 use fine-tuning or not? If yes, I’m wondering whether the same fine-tuning strategy (i.e., entwined) is adopted for all the candidates?
8. In Table 6, only the accuracy of SInGE when 90% of parameters removed is provided but not the other methods. It would be appreciated if the authors could provide the performance of other methods at this setting to support the superiority of SInGE.

**Limitations:**

The authors have included adequate limitations.

**Strengths And Weaknesses:**

## Strength
1. Motivation is very clear and the idea of using the approximated integration as the criterion of pruning is a good way to measure the influence of forcing a parameter to 0.
2. This paper has clear writing and easy to follow.

## Weaknesses

1. Figure 1 is not effective and clear enough to facilitate the readers’ understanding on the arguments made in Sec 3.1. One reason is that the notation $\mu$ is not introduced before its appearance. Also, the claims at line 129 – 133 about the effects of the change in values are not clearly depicted in Figure 1.

2. The claim that SInGE significantly outperforms existing state-of-the-art DNN pruning methods seems to be a bit inappropriate as it’s sometimes inconsistent with what demonstrates in the empirical studies. For example, in Table 5, the accuracy of SinGE is nearly the same as those of Adapt-DCP, ManiDP-A and MDP. However, the latter remove significantly more parameters or FLOPS than SinGE does.

---

> ### Author Response · Authors · 2022-08-02
> **Answer to Reviewer kiMH**
>
> **Figure 1 is not effective and clear enough to facilitate the readers' understanding on the arguments made in Sec 3.1. One reason is that the notation is not introduced before its appearance. Also, the claims at line 129 -- 133 about the effects of the change in values are not
> clearly depicted in Figure 1.**
>
> The purpose of Figure 1 is to show that importance measurement are very local and don't offer good evaluations for drastic changes such as zeroing out. As such, by showing that the norm of the neurons as well as the norm of its gradients, it does show different scenarios that are not well handled when evaluating the importance only once.
>
> As for the example given by the reviewer, example (b) in Figure 1, depicts a case where driving the norm towards zero causes an increase of the gradient norm, thus globally increasing its pruning cost (as the rectangle areas increase) making it a bad candidate for pruning.
> Conversely, in other cases such as (c) the higher initial cost keeps getting lower as the value of the neuron decreases. We updated the
> caption to better introduce the notation and make this easier to understand.
>
> **The claim that SInGE significantly outperforms existing state-of-the-art DNN pruning methods seems to be a bit inappropriate as
> it's sometimes inconsistent with what demonstrates in the empirical studies. For example, in Table 5, the accuracy of SinGE is nearly the
> same as those of Adapt-DCP, ManiDP-A and MDP. However, the latter remove significantly more parameters or FLOPS than SinGE does.**
>
> First, over the three main comparisons to the state-of-the-art (Table 3, 4 and 5), there is only one sub-category of one benchmark where
> state-of-the-art methods come close to ours, while SinGE performs significantly better elsewhere. Second, in this specific case, the reviewer claim that the accuracy are \"nearly the same\" but this is due
> to the already high accuracy in the sense that the original model has 71.8 accuracy and pruned models lie between 71.4 and 71.67. In such cases, even marginal absolute improvements are very significant relative improvement. Furthermore, such methods fail to remove more parameters and FLOPs while preserving decent accuracy (see Table 5 of the revised version/table 1 in the authors' response). We believe those results highlight the interest of the proposed approach.
>
> | goal |               Method              |  \% params rm  |   \% FLOPS rm   |               accuracy               |
> |:----:|:---------------------------------:|:--------------:|:---------------:|:------------------------------------:|
> |   -  |              baseline             |      0.00      |       0.00      |                 71.80                |
> | 30\% |          CBS (arxiv 2022)         |      30.00     |        -        |                 71.48                |
> |      |       Adapt-DCP (TPAMI 2021)      | **35.01**      |      30.67      |                 71.4                 |
> |      |        ManiDP-A (CVPR 2021)       |        -       |  **37.2**       |                 71.6                 |
> |      |               SInGE               |      30.96     |      31.54      |           **71.67** $\pm$ 0.06       |
> | 40\% |          CBS (arxiv 2022)         |      40.00     |        -        |                 69.37                |
> |      |          MDP (CVPR 2020)          | **43.15**      |        -        |                 69.58                |
> |      |               SInGE               |      40.90     |      42.30      |         **70.47** $\pm$ 0.09         |
> | 50\% |          CBS (arxiv 2022)         |      50.00     |        -        |                 62.96                |
> |      | Adapt-DCP (TPAMI 2021)            |  -             | 45.0            |                 64.13                |
> |      |  \revision{ManiDP-A (CVPR 2021)}  |  -             |       48.8      |                 69.62                |
> |      |         Accs (arxiv 2021)         |      50.00     |        -        |                 69.76                |
> |      |               SInGE               |    **50.13**   |    **48.90**    | **70.01** $\pm$ 0.22                 |
>
> [Table 1: Comparison with existing structured pruning methods on MobileNet V2 backbone for ImageNet]
>
> **As far as I undertand, the integration evaluation will take much time, so do the authors think that adding an experiment of the time efficiency necessary?**
>
> Indeed the proposed integrated gradient criterion comes at the expanse of an additional (linearly increasing with the number of integration steps) computational cost. To answer your concern, we propose a novel benchmark on Cifar10 where we work at a constant processing (pruning+fine-tuning) time: specifically, as we empirically measure that 1 integrated gradient step costs approximately half the time of a fine-tuning step, we modulate the numbers of fine-tuning steps to reach a similar processing time, and compare the accuracy of the pruned networks. Results can be found in Table 2.

---

> > ### Author Response · Authors · 2022-08-02
> > **Answer to Reviewer kiMH (2)**
> >
> > |                        method                        | pruning rate / FLOPs removed |          accuracy         |
> > |:----------------------------------------------------:|:----------------------------:|:-------------------------:|
> > |                  magnitude $C_{L^1}$                 |         86.46 / 85.00        |      69.42 $\pm$ 2.11     |
> > |                  magnitude $C_{L^2}$                 |         86.46 / 85.00        |      75.38 $\pm$ 2.76     |
> > |               gradients $C_{\nabla^2}$               |         86.46 / 85.00        |      79.55 $\pm$ 1.97     |
> > |    magnitude $\times$ grad $C_{L^2\times\nabla^2}$   |         86.46 / 85.00        |      91.05 $\pm$ 0.03     |
> > |        integrated gradients $C_{\text{SG}^2}$        |         86.46 / 85.00        |      92.01 $\pm$ 0.13     |
> > | integrated magnitude $\times$ grad $C_{\text{IG}^2}$ |         86.46 / 85.00        |    **93.07** $\pm$ 0.06   |
> >
> > [Table 2: Comparison of the performance of the different pruning criterion on a ResNet 56 for Cifar10 under the same time constraint]
> >
> > We believe these results highlight the interest of the proposed approach and added them in the revised version.
> >
> > **There are a few typos, for example, at the range of $\mu$ at line 138, the spellings of trade-offs at line 167, and the incompleteness of parentheses at line 181.**
> >
> > We would like to thank the reviewer for pointing-out typos which we corrected in the revised version of the article. However, we would like to clarify that there is no mistake in the range of parameter $\mu$ as it should be non-zero (otherwise the weights would be immediately pruned) nor equal to one as the integral would be stationary.
> >
> > **As I suppose the bolded numbers in the tables stand for the best results, some of those in Table 3 and 4 are not correctly labeled, which is a bit misleading.**
> >
> > We disagree with the reviewer on the labeling in Table 3. In Table 3, we compare the method to structured pruning techniques thus consider only structured pruning techniques for labeling, other methods are just here as reference. As for Table 4, The only contest would be for RED in terms of accuracy with 76.1 while we achieve 76.05. We corrected this in the revised version.
> >
> > |  top1 accuracy | pruning method | structured |        \% parameters removed        |
> > |:--------------:|:--------------:|:----------:|:-----------------------------------:|
> > | 91.5 $\pm$ 0.1 |       RED      |   Yes   |                 85.0                |
> > |                |       LP       |   Yes   |                 84.0                |
> > |                |       LP       |   No    |                 92.6                |
> > |                |       LDI      |   Yes   |                  88                 |
> > |                |       DPF      |   No    |                 90.0                |
> > |                |       HAP      |   Yes   |                 90.0                |
> > |                |  SInGE (ours)  |   Yes   | **91.3** $\pm$ 0.27                 |
> > | 93.5 $\pm$ 0.1 |       GDP      |   Yes   |                 65.6                |
> > |                |       HAP      |   Yes   |                 76.2                |
> > |                |  SInGE (ours)  |   Yes   | **84.3** $\pm$ 0.71                 |
> >
> > [Table 3: State-of-the-art pruning methods performance on ResNet 56 on Cifar10.]

---

> > > ### Author Response · Authors · 2022-08-02
> > > **Answer to Reviewer kiMH (3)**
> > >
> > > |       Method       |             \% params rm             |              \% FLOPS rm             |               accuracy               |
> > > |:------------------:|:------------------------------------:|:------------------------------------:|:------------------------------------:|
> > > |      baseline      |                 0.00                 |                 0.00                 |                 76.15                |
> > > |  Hrank (CVPR 2020) |                 36.67                |                 43.77                |                 74.98                |
> > > | RED (NeurIPS 2021) |                 39.6                 |                 42.7                 |                 76.1                 |
> > > |   HAP (WACV 2022)  |                 44.59                |                 33.82                |                 75.12                |
> > > | SRR-GR (CVPR 2021) |                   -                  |                  45                  |                 75.76                |
> > > |  SOSP (ICLR 2021)  |                  49                  |                  45                  |                 75.21                |
> > > | SRR-GR (CVPR 2021) |                   -                  |                  55                  |                 75.11                |
> > > |        SInGE       | **50.80** $\pm$ 0.02                 | **57.35** $\pm$ 0.11                 | **76.05** $\pm$ 0.07                 |
> > > | RED (NeurIPS 2021) |                 54.7                 |                 55.0                 |                 71.1                 |
> > > |  SOSP (ICLR 2021)  |                  54                  |                  51                  |                 74.4                 |
> > > |   GDP (ICCV 2021)  |                   -                  |                  55                  |                 73.6                 |
> > > |   HAP (WACV 2022)  |            **65.26**                 |                 59.56                |                 74.0                 |
> > > | OTO (NeurIPS 2021) |                 64.1                 |                 65.2                 |                 73.3                 |
> > > |        SInGE       |      63.78 $\pm$ 0.01                | **65.96** $\pm$ 0.21                 |   **74.7** $\pm$ 0.31                |
> > >
> > > [Table 4: Comparison between existing structured pruning performance on ResNet 50 on ImageNet. In both the low ($<50\%$ parameters removed) and high ($>50\%$) pruning regimes, SInGE achieves remarkable results.]
> > >
> > > **The top-1 accuracy of integrated magnitude method without fine-tuning and with 1000-step entwined fine-tuning seems to be the same (both reported as 85.38). I'm wondering why there is no increase in the accuracy.**
> > >
> > > This is a mistake on our part, we updated the results which don't actually change the conclusions. We double checked every other Tables to ensure such errors wouldn't happen an other time.
> > >
> > > **May I ask what does the notation "1\<" stand for in Table 1?**
> > >
> > > This indicates standard deviations lower than $1$ for accuracies at chance level.
> > >
> > > **As the performance of SinGE is very close to its counterpart in many cases, could the authors please provide the error bars for the results reported in Table 2-6?**
> > >
> > > To answer your concern we added the standard deviations to the tables in the revised paper. We believe these values show the stability of our results as well as the statistical significance and quality of the proposed results.
> > >
> > > **May I ask whether the results in Table 3 use fine-tuning or not? If yes, I'm wondering whether the same fine-tuning strategy (i.e., entwined) is adopted for all the candidates?**
> > >
> > > As the proposed entwined pruning (via integrated gradient criterion) and fine-tuning is a contribution of this work, state-of-the-art methods do
> > > not use this approach. However, please note that 1) a comparison between entwined and post-pruning fine-tuning can be found in Table 2, and 2) in our work we only perform 50k fine-tuning steps total whereas most papers (including e.g. Adapt-DCP, ManiDP-A and MDP) use $\approx 1M$ updates. Once again, we belive that this shows the quality of our results.
> > >
> > > **In Table 6, only the accuracy of SInGE when 90% of parameters removed is provided but not the other methods. It would be appreciated if the authors could provide the performance of other methods at this setting to support the superiority of SInGE.**
> > >
> > > Unfortunately, to the best of our knowledge, no other paper in the unsupervised learning literature report those values for comparison. However, we believe that, as such, Table 6 shows that our approach, while in principle being designed for structured pruning, can be straightforwardly applied to unsupervised pruning and achieve competitive results without bells and whistles.

---

> > > > ### Comment · Reviewer_kiMH · 2022-08-08
> > > > **Thanks for the response**
> > > >
> > > > I think my concerns has been properly addressed. And based on this I would like to reconsider my rating, especially due to the novelty of the integration method.

---

### Meta-Review · Area_Chair_fSF7 · 2022-08-31

**Recommendation:** Accept
**Confidence:** Certain

**Metareview:**

Novel pruning method based on integrated gradients. Reviewers agreed that the method is well-motivated and that the comparisons showcase the potential of this method. There are some concerns regarding fairness of the comparisons in terms of flops and parameter count. I believe some of the rebuttal answers from the authors address those concerns. I think this work is novel and interesting enough to be accepted at NeurIPS.

**Award:**

No

---

### Decision · Program_Chairs · 2022-09-14

Accept